# Adaptive Mechanisms of Halophytes and Their Potential in Improving Salinity Tolerance in Plants

**DOI:** 10.3390/ijms221910733

**Published:** 2021-10-03

**Authors:** Md. Mezanur Rahman, Mohammad Golam Mostofa, Sanjida Sultana Keya, Md. Nurealam Siddiqui, Md. Mesbah Uddin Ansary, Ashim Kumar Das, Md. Abiar Rahman, Lam Son-Phan Tran

**Affiliations:** 1Department of Plant and Soil Science, Institute of Genomics for Crop Abiotic Stress Tolerance, Texas Tech University, Lubbock, TX 79409, USA; mdmerahm@ttu.edu (M.M.R.); skeya@ttu.edu (S.S.K.); 2Department of Biochemistry and Molecular Biology, Bangabandhu Sheikh Mujibur Rahman Agricultural University, Gazipur 1706, Bangladesh; nuralambmb@bsmrau.edu.bd; 3Department of Biochemistry and Molecular Biology, Jahangirnagar University, Savar, Dhaka 1342, Bangladesh; ansarymu.ju@gmail.com; 4Department of Agroforestry and Environment, Bangabandhu Sheikh Mujibur Rahman Agricultural University, Gazipur 1706, Bangladesh; ashimbsmrau@gmail.com (A.K.D.); abiar@bsmrau.edu.bd (M.A.R.); 5Institute of Research and Development, Duy Tan University, Da Nang 550000, Vietnam

**Keywords:** coastal areas, halophytes, phytoremediation, salt tolerance mechanisms, soil salinity, transgenic plants

## Abstract

Soil salinization, which is aggravated by climate change and inappropriate anthropogenic activities, has emerged as a serious environmental problem, threatening sustainable agriculture and future food security. Although there has been considerable progress in developing crop varieties by introducing salt tolerance-associated traits, most crop cultivars grown in saline soils still exhibit a decline in yield, necessitating the search for alternatives. Halophytes, with their intrinsic salt tolerance characteristics, are known to have great potential in rehabilitating salt-contaminated soils to support plant growth in saline soils by employing various strategies, including phytoremediation. In addition, the recent identification and characterization of salt tolerance-related genes encoding signaling components from halophytes, which are naturally grown under high salinity, have paved the way for the development of transgenic crops with improved salt tolerance. In this review, we aim to provide a comprehensive update on salinity-induced negative effects on soils and plants, including alterations of physicochemical properties in soils, and changes in physiological and biochemical processes and ion disparities in plants. We also review the physiological and biochemical adaptation strategies that help halophytes grow and survive in salinity-affected areas. Furthermore, we illustrate the halophyte-mediated phytoremediation process in salinity-affected areas, as well as their potential impacts on soil properties. Importantly, based on the recent findings on salt tolerance mechanisms in halophytes, we also comprehensively discuss the potential of improving salt tolerance in crop plants by introducing candidate genes related to antiporters, ion transporters, antioxidants, and defense proteins from halophytes for conserving sustainable agriculture in salinity-prone areas.

## 1. Introduction

Land degradation resulting from salinity puts global agriculture in jeopardy, as it engulfs 3600 million hectares (Mha) out of 5200 Mha of world agricultural land, leading to the loss of USD 27.5 billion every year [1,2]. Furthermore, reports anticipate that the world population will increase to 9.7 billion by 2050, which would require 70% greater food production by that time, exerting additional pressure on the shrinking areas of agricultural land [3]. To feed the growing human population, many parts of the world in peril of drought are still using salt-contaminated ground water for irrigation [4]. Inappropriate agricultural management practices, such as improper drainage, also exacerbate soil salinity, making the soil untenable for agriculture, as evident in Punjab of Pakistan and Fubei of China [5]. Furthermore, climate-driven impacts, such as rising sea levels, tidal changes, drifting seawater droplets, and disastrous events, such as tsunamis, cause accumulations of large pools of concentrated seawater on agricultural land [6]. Consequently, salinity threatens around 1.5 Mha of land each year, which will increase the risk of losing 50% of cultivable lands by the middle of the 21st century [7,8].

The Harmonized World Soil Database (HWSD) estimated that around 1130 Mha of land is inclined to the incursion of varying salinity levels. Major categories of salinity-affected soils are saline, sodic, and saline-sodic, accounting for 60, 26, and 14%, respectively, of all salinity-affected soils [9] (Figure 1A). Approximately, 736 (65%), 227 (20%), 53 (5%), and 114 (10%) Mha of all salinity-affected soils are further categorized as slightly, moderately, highly, and extremely salinity-affected soils, respectively [9] (Figure 1B). The extent and severity of salinity-affected soils vary by regions across the world. The largest salinity-affected areas are in the Middle East, Australia, North Africa, and the former Soviet Union, which account for 176, 169, 161, and 126 Mha of land, respectively [9] (Figure 1C).

Plants grown in salinity-affected areas accumulate higher levels of toxic ions responsible for different physiological abnormalities, including ionic imbalance, impaired gas exchange performance, loss of water homeostasis, alterations in the levels of metabolites, and reactive oxygen species (ROS)-mediated oxidative damage to the cellular compartments [10,11]. To counteract the detrimental effects of salt stress, plants adopt some fundamental mechanisms, including (i) toxic ion (Na^+^ and Cl^−^) exclusions or their compartmentation into vacuoles or old tissues; (ii) compatible solute accumulations; (iii) synthesis of numerous stress adaptation-related endogenous metabolites; and (iv) synthesis and activation of antioxidant enzymes [e.g., catalase, (CAT), superoxide dismutase (SOD), ascorbate peroxidase (APX), glutathione peroxidase (GPX), peroxidase (POD), and glutathione *S*-transferase (GST)], and non-enzymatic antioxidants [e.g., carotenoids, phenolic compounds, flavonoids, ascorbate (AsA), and glutathione (GSH)] [10,12,13,14]. Unfortunately, the traditional crops cannot deploy salt tolerance potential to survive in salinity-affected soils, and a wide range of genetic diversity for salt tolerance in conventional crops, even local landraces, remains elusive. Interestingly, some plants, such as halophytes, comprising only <2% of the world’s flora, can sustain and thrive under extreme salinity through the adoption of intrinsic mechanisms related to selective ion uptake, Na^+^ compartmentalization, ion homeostasis, production of defense metabolites, salt gland and salt bladder formations, and improved antioxidant properties [14,15]. Given their ability to tolerate high salinity, the cultivation of halophytes on salt-contaminated soils has been introduced as an alternative solution to economical desalination and restoration of highly degraded saline and sodic soils through the process, called phytoremediation [16]. It is worth noting that molecular analyses, such as transcriptome sequencing, have been widely used to investigate the gene expression dynamics of numerous species in response to salt stress [17]. Nevertheless, exploration of salt tolerance mechanisms of halophyte species with the final goal of transferring their salt tolerance-associated traits and/or genes to traditional crops is timely and imperative, which may also contribute to ensuring sustainable agriculture in salinity-affected areas.

Considering the above facts, the present review focuses on the latest update of the adverse effects of salinity both on soils and plants, as well as the intrinsic physiological, biochemical, and molecular mechanisms that halophytes employ to adapt under salinity. In addition, mechanistic aspects of desalinization and phytoremediation potential of halophytes for improving agricultural output from salt-contaminated soils will also be appraised. Finally, a comprehensive update on transcriptomic studies in halophytes and the genetic potential of halophytes for enhancing salt tolerance in traditional crops will be discussed by considering the genes associated with salt tolerance-related traits of halophytes.

## 2. Adverse Effects of Salinity on Soil Properties

Salinity causes immense alterations in soil properties. For instance, high salt concentrations in soils may result in high bulk density, soil compaction, poor soil structure, clay dispersion, and surface crusting [18]. In addition, salinity-induced poor hydraulic conductivity can severely impede water infiltration, leading to poor growth performance and yield reduction in plants [19]. Accumulation of salt in soils also has detrimental effects on soil microbial communities and their enzymatic activities [20]. Because of their increased sensitivity to environmental perturbations, soil microbiomes and associated enzyme activities are regarded as early indicators for measuring the degree of soil degradation and changes in soil quality [21]. Microbial activities play critical roles in the decomposition of soil organic matter (OM), mineralization of soil, and stabilization of soil aggregates [22,23]. Salinity-induced osmotic imbalance can lead to water leakage from the microbial cells and, consequently, plasmolysis and death of microorganisms [24]. On the other hand, microorganisms may live in low osmotic potential environments by accumulating osmolytes, which is an energy-intensive process [25]. Soil enzymes, including ureases, dehydrogenases, CATs, alkaline phosphatases, and *β*-glucosidases, derived from soil microbes, plant root exudates, and plant and animal residues, play central roles in the regulation of all the processes involved in OM decomposition, as well as in the cycling of soil nutrients [26]. Despite their immense influence on the alteration of soil quality, only a few studies were conducted to assess the potentiality of these enzymes in changing soil quality in coastal areas. In addition, comprehensive information on how salinity influences soil quality is still ambiguous due to the intricate nature of soils [25]. Therefore, future research should focus on unraveling their intrinsic processes in a spatio-temporal manner.

## 3. Mechanisms of Salinity-Mediated Adverse Effects on Plants

Elevated soil salinity can hinder plant performance in several ways. Salinity-induced water paucity severely affects seed germination by lowering the osmotic potential of the soil water, which interrupts imbibition in plant seeds [27,28]. Any disruption in seed imbibition can alter enzyme activities and protein metabolism, perturb hormonal balance, retard seed reserve utilization, and very often distort the ultrastructure of cells, tissues, and organs [29,30,31]. Reduction of plant growth and biomass after germination is the foremost negative impact of soil salinity, which depends on salinity levels, duration of salt exposure and type of plant species [27]. Plant cells can lose water after plant exposure to salinity within a few seconds, and as a result, cell shrinkage occurs [32]. Cells regain their original volume over hours, but cell elongation rates are reduced [33]. Over days, cell division rates are also severely reduced, leading to decreased plant growth [34]. Salinity can also cause visible injuries in the oldest leaves, the demise of older and younger leaves, which may result in death of the plants before seed maturation [32,33]. Nevertheless, the reduced growth rate of plants under salinity is predominantly regulated by root responses, as roots are directly exposed to saline soils, and act as a key conduit for mineral translocation to aerial parts of the plant [34]. As mentioned above, salt stress lowers the water potential of soil solution, which also interrupts water conductivity of roots, ultimately leading to a reduction in water and mineral influx to the plant body [35].

High concentrations of Na^+^ and Cl^−^ in soil solution restrict uptake and accumulation of macronutrients like Ca^2+^ and Mg^2+^, which perturbs signal transduction, membrane stability, chlorophyll synthesis, and stomatal opening [34,36]. Furthermore, salinity’s adverse effects on photosynthesis are intricately associated to the destruction of photosynthetic pigments, and the underlying facts are still not adequately elucidated. Exposure of plants to salt stress causes a decrease in stomatal density, stomatal conductance, photosynthetic activity, and carbon assimilation, but an increase in mesophyll resistance, thereby reducing light absorption efficiency of photosystem (PS)-I and PS-II [37]. Salt stress also suppresses the activity of ribulose-1,5-bisphosphate carboxylase/oxygenase (RuBisCO), a key enzyme involved in carbon dioxide (CO_2_) assimilation during the dark phase of photosynthesis [38]. In addition, salinity also induces the production of ROS, including hydroxyl radical (OH^•^), superoxide (O_2_^•−^), hydrogen peroxide (H_2_O_2_) and singlet oxygen (^1^O_2_), in cellular organelles like chloroplasts, mitochondria, peroxisomes, and endoplasmic reticulum [12,13]. The elevated levels of ROS can negatively affect plant growth and developmental processes by causing protein synthesis reduction, cell membrane destruction, genomic instability, and damage to photosynthetic apparatus [13,39]. Although Cl^−^ is the most abundant anion in saline soils, its effects on plant performance, as well as its uptake and transport mechanisms within plants have still received less attention than those of Na^+^ [40]. As an essential micronutrient, Cl^−^ plays a putative role in enzyme activation in the cytoplasm, oxygen evolution during photosynthesis, stabilization of membrane potential, and maintenance of turgor pressure and pH of the plant cells [40]. However, once excessively accumulated, Cl^−^ has even more detrimental effects than Na^+^ in destabilizing the normal functions of the cells [40]. All above-mentioned salinity-induced adverse effects on plants trigger diminution of crop yield, although the intensity and damage vary depending on plant species.

## 4. Halophytes and Their Classification

Halophytes are plants that have the ability to survive and complete their life cycles in environments with high salinity, without suffering major negative impacts on their growth or development [15]. Based on their levels of tolerance to different types of saline soils, halophytes can broadly be categorized into two groups: obligate halophytes and facultative halophytes. Obligate halophytes can sustain their sufficient growth and development in high saline habitats containing salinity levels similar to seawater [41]. On the other hand, facultative halophytes are generally found in low saline habitats, although they can thrive in salt-devoid environments as well [42]. Additionally, Von Sengbusch [43] used the eco-physiological characteristics to distinguish obligate and facultative halophytes from habitat-independent halophytes. Habitat-independent halophytes are not necessarily native to saline habitats; they usually prefer to grow in salt-free soils but can cope with the saline conditions [43].

Nevertheless, the comprehensive classification of halophytes is still ambiguous because scientists defined halophytes by considering different aspects [44], including integration of anatomical observations, intensity of salinity and mechanisms of salt tolerance [45,46]. More recently, Santos et al. [47] developed a database of halophytes, namely eHALOPH (http://www.sussex.ac.uk/affiliates/halophytes/; accessed on 20 July 2021), which accumulates the information of plant species that can survive under 80 mM or higher salt concentrations. The authors classified the halophytes into seven categories: hydro-halophytes, chasmophytes, phreatophytes, psammophiles, xerophytes, weedy halophytes, and xero-halophytes. Hydro-halophytes typically grow in aquatic conditions or on wet soil; chasmophytes are found on cliff-tops, rocky and sandy seashores, and saltmarshes; phreatophytes are deep-rooted plants obtaining water from a deep underground source that may or may not be saline; psammophiles are usually found in sandy soils; xerophytes are adapted to extreme drought-prone areas; weedy halophyte species are predominantly invading and colonize highly disturbed sites or regions; and xero-halophytes are adapted to inland salt desert and saline habitats [47].

In addition, based on the physiological basis of salt tolerance as well as the accumulation and transportation of ions, Breckle et al. [48] classified halophytes into three major categories, including recretohalophytes, euhalophytes, and salt-exclusion halophytes (also referred as salt excluders). Recretohalophytes have unique salt-secreting structures, such as salt glands and salt bladders, whereas euhalophytes have the ability to dilute absorbed salts in their succulent leaves or stems. Salt excluders, on the other hand, can prevent the uptake of salt ions from soils or shed leaves containing toxic levels of salt [49,50]. With these complex classifications, it is important to acquire knowledge on their extent of salt tolerance levels, and what the intrinsic mechanisms they adopt to combat salinity effects.

## 5. Physiological Mechanisms Associated with Halophyte Adaptation to Soil Salinity

Comprehensive research on halophytic plants is crucial for a better understanding of their salt tolerance mechanisms and their exploration for breeding purposes. To withstand and continue to be grown well in a saline environment, halophytic plants employ different morphophysiological strategies, such as salt exclusion, salt excretion through specialized organs, i.e., salt glands and/or salt bladders, and dilution of salt ions by succulence. We will review these morphophysiological mechanisms in light of their functions in halophytes’ resistance to salt stress (Figure 2 and Figure 3; Table 1).

### 5.1. Salt Exclusion

Halophytic plants, particularly hydro-halophytes and some phreatophytes, were reported to exclude excessive salts by root system-induced ultrafiltration mechanisms. More specifically, installing apoplastic barriers at the roots improves bypass flow resistance, resulting in effective exclusion of salts from the roots and, as a result, a reduced accumulation of toxic ions in the aboveground shoots through the transpiration stream [51] (Figure 2; Table 1). The root endodermis and exodermis are two distinct cell barrier layers with highly specialized roles ascribed to two particular cell wall features: (a) the Casparian band (CB) and (b) the suberin lamella (SL) (Figure 2). The CB, a paracellular lignin deposition in the endodermis and exodermis cell walls, forces all apoplastic transport into the strictly regulated symplastic system [52,53]. Furthermore, SL, primarily made of long-chain fatty acids, permeates the whole exodermis/endodermis cell wall, providing a hydrophobic barrier that aids in ion and water uptake regulation [54,55]. SL is thought to play an essential role in preventing the direct movement of water and ions from the apoplast into the endodermal protoplasts [56,57]. It was recently discovered that SL acts as an apoplastic barrier at the site of lateral root emergence where CBs have not yet developed [57]. A limited number of studies have been performed on halophytes and shown that apoplastic barriers are crucial for salt adaptation of *Suaeda maritima* [58], *Avicennia officinalis* [59], *S. salsa* [60], and *A. marina* [61]. However, in comparison with CBs, there is less evidence for the role of SL in plant salt tolerance; and hence, more research is needed to assess the role of SL in ion exclusion from plants under salinity.

Importantly, salt overly sensitive 1 (SOS1) in the cortical cells plays an essential role in exclusion of Na^+^ from roots (Figure 2). Recently, Liu et al. [62] reported that *S. salsa* exhibited greater expression of *S. salsa SOS1* (*SsSOS1*) in roots of intertidal *S. salsa* population than in those of inland *S. salsa* population, which was correlated with the increased exclusion of Na^+^ by roots; and thus, it could thrive in a high saline environment (Table 1). Similarly, greater expression of *Karelinia caspia SOS1* (*KcSOS1*) gene in the roots of the recretohalophyte *K. caspia* was also observed [63] (Table 1). However, the mechanisms underlying root system-induced ultrafiltration under saline conditions remain elusive and require further in-depth investigations. Many reports have recommended that salt exclusion mechanism as a cure to reduce accumulations of toxic levels of Na^+^ and Cl^−^ in plant bodies [64]. A strong argument could, however, be raised against this mechanism since it may elevate soil salinity levels by restriction of salt uptake by roots [65].

### 5.2. Salt Excretion

Halophytes, particularly recretohalophytes, directly secrete remarkably higher contents of salt ions onto the leaf surface, for dealing with maintenance of cellular ion homeostasis, through a unique epidermal structure called salt glands, making them superior compared with other classes of halophytes and non-halophytes [50] (Figure 3). Several species of the *Limonium* genus, including *L. bicolor*, *L. aureum*, *L. gmelinii*, *L. otolepis,* and *L. sinuatum*, are good examples of recretohalophytes, which boost salt tolerance by enhancing salt secretion via the salt glands [66,67,68] (Table 1). Species of another recretohalophyte genus, *Tamarix*, are salt-tolerant by increasing the density of salt glands and the rate of salt secretion per salt gland [69] (Table 1). However, salt glands in different halophytes species possess diverse structural characteristics, and the number of cells that make up salt glands are different [50]. Still, their common features include the followings: (i) salt glands are surrounded by a thickened cuticle; (ii) having many plasmodesmata between the cells, a large number of highly developed mitochondria and many small vesicles in the cytoplasm; and (iii) having no chloroplasts in the cells [50,70].

Multi-cellular salt glands varying from 4 to 40 cells are found in dicotyledonous recretohalophytes, including *L. bicolor* [66], whereas bi-cellular salt glands are prominent in monocotyledonous recretohalophytes of the Poaceae family [71]. Ions secreted through salt glands include various cations (Na^+^, K^+^, Ca^2+^, Mg^2+^, Fe^2+^, Mn^2+^, and Zn^2+^) and anions (Cl^−^, Br^−^, I^−^, SO_4_^2−^, PO_4_^3−^, and NO_3_^−^), depending largely on the environment [50]. The actual pathways of salt secretion from salt glands remain to be determined, although several possible pathways based on recent studies have been suggested by Yuan et al. [70] and Lu et al. [50]. Nevertheless, it was also demonstrated that salt glands might impose detrimental threats to plants by causing leaf dehydration because the glands produce salt crystals on the leaf surface [65]. This problem can be minimized through *de novo* synthesis of compatible solutes, but it requires high energy, i.e., 30 to 109 molecules of ATP to produce one molecule of any compatible solute, which ultimately leads to a penalty in crop yield [65].

**Figure 2 ijms-22-10733-f002:**
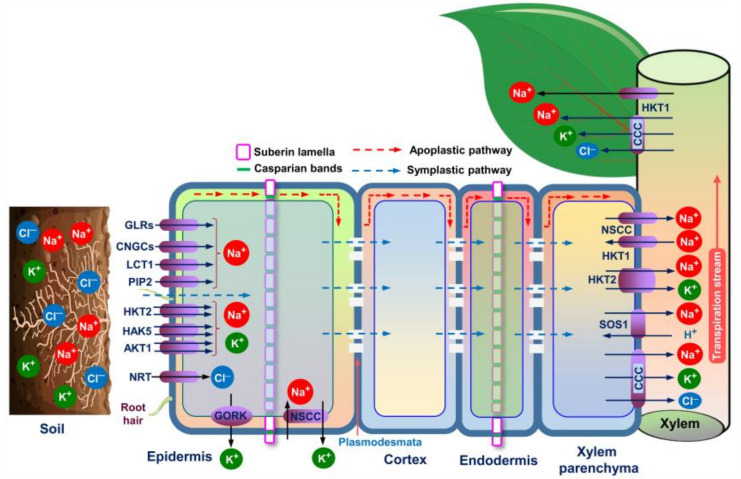
Putative salt uptake and accumulation mechanisms in an ideal halophyte plant (modified from Zhao et al. [11], Arif et al. [34], and Arbelet-Bonnin et al. [72]. Na^+^ is transported from soils to/within plant root cells via two major pathways: symplastic and apoplastic. The major transporters/channels embedded in the root epidermis for up-taking Na^+^ are GLRs, CNGCs, LCT1, PIP2, HKT2, HAK5, and AKT1. In the presence of Na^+^, K^+^ is competitively uptaken via HKT2, HAK5, and AKT1. The entrance of Cl^−^ into the root cells is mediated by separate transporters, including NRT. NSCC and GORK channels, on the other hand, are involved in salinity-induced K^+^ exclusion from the root epidermis. Apoplastic barriers, consisting of Casparian bands and suberin lamellae in the epidermis and endodermis of the roots, can block and/or limit the bypass flow of Na^+^ and other solutes moving to the aboveground parts. Passive Na^+^ entry into the xylem is facilitated by NSCCs, while cotransporters like SOS1, CCC, and HKT2 are required for active loading of Na^+^ into the xylem. Na^+^ can return to xylem parenchyma using HKT1. The xylem-loaded Na^+^, K^+^ and Cl^−^ can be transported up through the transpiration stream, and subsequently enter into the aboveground organs via CCC and HKT1. Abbreviations: AKT1, *Arabidopsis* K^+^ transporter 1 (a shaker-type K^+^ channel); CCC, cation-chloride cotransporter; CNGC, cyclic nucleotide-gated channel; GORK, gated outwardly rectifying K^+^ channel; GLR, glutamate receptor-like channel; HKT, high-affinity potassium transporter; HAK, high-affinity potassium uptake transporter; LCT, low-affinity cation transporter; NRT, nitrate transporter; NSCC, non-selective cation channel; PIP2, plasma membrane intrinsic protein; SOS1, salt overly sensitive 1.

Epidermal bladder cells (EBCs), also known as salt bladders, can be found in more than half of the halophytes [65,73]. The EBC have a modified non-glandular trichome structure that is divided into an apical and basal cell, and the apical cell gains a globular shape, typically with a cell volume of 500 nL and an average diameter of 1 mm [65,73,74] (Figure 3). The diameter of EBCs often stretches ~10× larger in comparison with normal epidermal cells to sequestrate excessive salts, by as much as 1000× more than the traditional leaf cell vacuoles can do [65].

**Figure 3 ijms-22-10733-f003:**
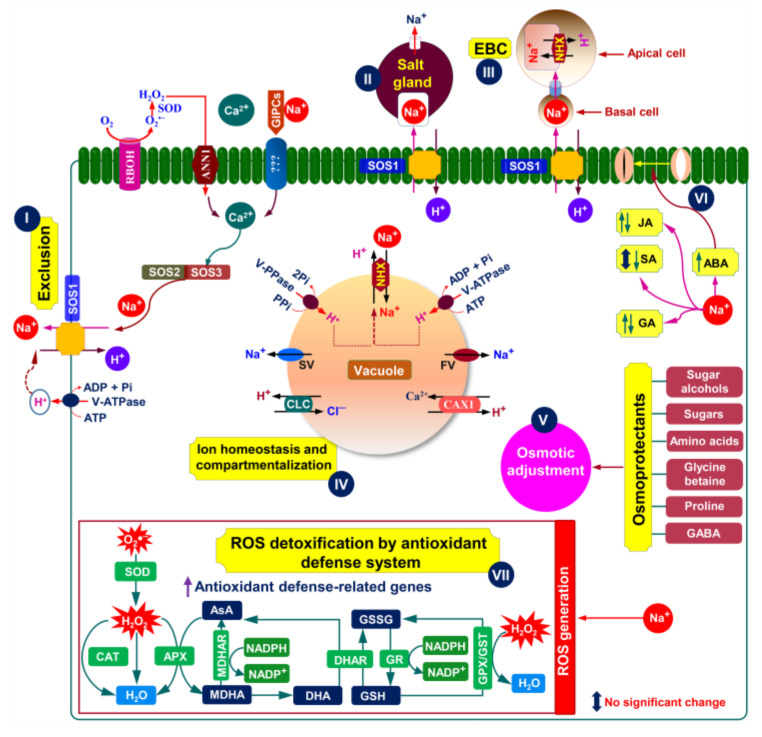
Putative salt tolerance mechanisms in an ideal halophyte plant (modified from Zhao et al. [11], Arif et al. [34], Arbelet-Bonnin et al. [72], Tran et al. [75], and Himabindu et al. [76]. The absorption of Na^+^ in plant cells is counterbalanced by active Na^+^ extrusion through SOS1 Na^+^/H^+^ exchanger (**I**). The RBOH-mediated ROS formation in the plasma membrane can activate ANN1-induced Ca^2+^-signaling pathway (**I**). The Na^+^ bound GIPCs can trigger Ca^2+^ influx through an unknown Ca^2+^ channel (**I**). Ca^2+^ influx mediated by either ANN1 or GIPCs is essential for activating the SOS-signaling pathway (**I**). The SOS3 relays salt-triggered Ca^2+^ signals to SOS2 kinase, forming the SOS3-SOS2 protein complex that phosphorylates SOS1 to induce Na^+^ efflux (**I**). To maintain ion homeostasis, halophytes can also sequestrate excess Na^+^ into the vacuoles and/or excrete Na^+^ via two unique structures, namely salt glands and salt bladders (**II** and **III**). It is also possible to avoid excessive Na^+^ accumulation in the cytosol by sequestrating it into the vacuoles (**IV**). Predominantly, Na^+^-sequestration in the vacuoles is conferred by tonoplast-based Na^+^/H^+^ exchangers of the NHX family (e.g., NHX1 or NHX2), fueled by either H^+^-PPase or H^+^-ATPase pumps (**IV**). Another equally important mechanism of vacuolar Na^+^-sequestration is the efficient control of tonoplast slow vacuolar (SV) and fast vacuolar (FV) activating ion channels, which may allow vacuolar Na^+^ to leak back into the cytosol (**IV**). Once, if both channels are downregulated or blocked by choline, Na^+^ leakage is inhibited, resulting in an increased level of salt tolerance (**IV**). On the other hand, Cl^−^ influx in the vacuoles is mediated by CLC (**IV**). To maintain cytosolic Ca^2+^ homeostasis under saline conditions, the influx of Ca^2+^ into the vacuole is mediated by CAX1 (**IV**). In response to salt stress, halophytes often accumulate various osmoprotectants, including proline, total free amino acids, sugars, polyols, glycine betaine, and GABA, which play vital roles in conferring an efficient osmotic adjustment (**V**). In addition, changes in the levels of different phytohormones may also help in stress adaptation; for instance, increased ABA level aids in controlling stomatal closure to maintain better water status at the tissue level (**VI**). It is worth mentioning that changes in the levels of phytohormones (e.g., JA, SA, and GA) in halophytes under salt stress varies dependently on the type of halophytes (**VI**). Furthermore, excessive salt accumulation can trigger oxidative stress via overproduction of ROS, such as O_2_^•–^ and H_2_O_2_ (**VII**). To fight against ROS-mediated oxidative damage, halophytes induce a vibrant antioxidant defense system by activating both enzymatic (SOD, CAT, APX, MDHAR, DHAR, GR, GPX, and GST) and non-enzymatic (AsA and GSH) antioxidants (**VII**). Abbreviations: ABA, abscisic acid; ANN1, annexin1; APX, ascorbate peroxidase; AsA, ascorbic acid; CAT, catalase; CAX, Ca^2+^/H^+^ exchanger; CLC, chloride channel (H^+^/Cl^–^ antiporter); DHA, dehydroascorbate; DHAR, dehydroascorbate reductase; EBC, epidermal bladder cell; GA, gibberellic acid; GIPC, glycosyl inositol phosphoryl ceramide; GST, glutathione *S*-transferase; GPX, glutathione peroxidase; GSH, glutathione; GSSG, oxidized glutathione; GR, glutathione reductase; GABA, gamma amino butyric acid; H_2_O_2_, hydrogen peroxide; JA, jasmonic acid; MDHA, monodehydroascorbate; MDHAR, monodehydroascorbate reductase; NHX, Na^+^/H^+^ exchanger; O_2_^•–^, superoxide; RBOH, respiratory burst oxidase homolog; ROS, reactive oxygen species; SOS, salt overly sensitive system; SA, salicylic acid; SOD, superoxide dismutase; V-PPase, vacuolar pyrophosphatase; V-ATPase, vacuolar H^+^ ATPase.

More fundamentally, EBCs have long been considered imperative for maintaining low Na^+^ levels in leaves, especially in young leaves [77]. Because younger leaves have small and underdeveloped vacuoles in the mesophyll cells, they rely solely on EBCs for internal salt sequestration [77]. Besides serving as a potential hotspot for storage of excessive Na^+^ and Cl^−^, EBCs may also store different metabolic compounds, ROS-scavenging metabolites and organic osmoprotectants [78]. A positive correlation between the presence and/or the number of EBCs and salt tolerance was reported in *Chenopodium quinoa* [74,78]. Some candidate genes related to EBC formation were recently identified using transcriptome analysis in *C. quinoa*, *Mesembryanthemum crystallinum,* and *Atriplex centralasiatica* species [79,80,81] (Table 1). More fundamentally, Imamura et al. [82] first identified a gene, named *reduced epidermal bladder cells* (*REBC*), involved in the EBC development in *C. quinoa*, laying the foundation for elucidating the EBC-associated stress tolerance mechanism in halophytes.

### 5.3. Succulence

Halophytes also adopt succulence mechanisms for ion homeostasis and accumulation of osmoprotectants to maintain cell turgor pressure under saline conditions [83,84] (Table 1). Succulent plants, popularly referred to as halo-succulents, have been characterized by thick leaves and stems, increased size of mesophyll cells, smaller intercellular space, enhanced tissue water content and high turgor pressure [83]. Succulent leaves possess more and stretched-sized mitochondria, which presumes to provide the excess energy required for salt compartmentalization and sequestration [84]. A large number of halophytes, including *Salsola drummondii*, *Achras sapota,* and *Sarcocornia fruticosa,* were reported to have efficient succulent mechanisms to confer salt tolerance via accumulation of elevated salts in their leaves and stems while maintaining improved photosynthetic efficiency [85,86,87] (Table 1). However, there is still much to learn about the ‘nuts and bolts’ of succulence. Therefore, current research should focus on (i) how the complex structural-functional coordination regulates the physiological processes in succulent plants; (ii) transcriptomics and gene-editing techniques to provide insights into the functional biology of succulent plants; and (iii) genetic engineering to introduce succulence traits into non-succulent plants to enhance their resilience to salinity-caused environmental perturbations.

**Table 1 ijms-22-10733-t001:** List of selected halophyte species along with their adaptive mechanisms favorable for salt tolerance.

Sl. No.	Species Name	Maximum Salt Tolerance Limit (mM)	Salt Tolerance Mechanisms	References
**1**	*Haloxylon salicornicum*	400	Increased levels of organic acids, including malate, gluconate, lactate, oxalate, citrate, and glycerate for activating TCA cycle Enhanced levels of sugars and sugar alcohols like glucose, sucrose, and glycerol for osmotic adjustment Enhanced levels of ABA and reduced level of JA for water conservation and reduction of oxidative stress	[88]
**2**	*Scorzonera hieraciifolia*	600	Enhanced levels of proline and soluble sugars for osmotic adjustment Improved ascorbic acid and glutathione levels, and the activity of SOD, GPX, APX, CAT, and GR for protection against oxidative stress	[89]
**3**	*Salsola crassa*	300	Increased abundance of stress-responsive proteins [e.g., elongation factor 1-gamma, HSC70, HSC80, EF-TU, eIF4A, isocitrate dehydrogenase (NADP) and citrate synthase] for assisting the better operation of TCA cycle and glycolysis Increased activity of CAT and GPX for oxidative stress mitigation	[90]
**4**	*Aster tripolium*	600	Enhanced levels of ABA and sugar alcohols for improving water status	[91]
**5**	*Lycium humile*	750	Enhanced levels of ABA and proline for water conservation and osmotic adjustment Elongated water-storage parenchyma of leaf cells for Na^+^ accumulation	[92]
**6**	*Limonium bicolor*, *L. gmelinii*, *L. otolepis, L. aureum, L. sinuatum,* and *L. gmelinii*	100, 50, 300, 350, 400, and 420, respectively	High salt gland density for increasing salt secretion capacity Enhanced expression of the genes encoding ion transporters (*LbHTK1*, *LbSOS1*, *LbPMA*, and *LbNHX1*) and vesicular transport proteins (*LbVAMP721*, *LbVAP27*, and *LbVAMP12*)	[66,67,68,69]
	*Tamarix gallica*, *T. ramosissima,* and *T. laxa*	300		
**7**	*Chenopodium quinoa* and *Atriplex centralasiatica*	300–400	Increased vacuolar sequestration of salts Epidermal bladder cells for storage of metabolites and compatible solutes Improved volume of bulliform cells of salt bladders	[73,78,80,81]
**8**	*Nitraria sibirica*	200	Improved activities of VHA and vacuolar H^+^-PPase, elevated expression of *NsVHA*, *NsVP1* and *NsNHX1* genes for vacuolar sequestration of Na^+^ Improved expression of two-pore K^+^ (*NsTPK*) for cytosolic K^+^ homeostasis	[93]
**9**	*Mesembryanthemum crystallinum*	400	Improved expression of *McNRT1*, *McHKT1*, *McKmt1*, *McCCC1*, *McCLC1*, *McNHX1,* and *McVmac1* genes for ion homeostasis Elevated expression of proline biosynthesis (*McP5CS*) and ononitol (*Mclmt1*) genes for osmotic adjustment	[75]
**10**	*Lobularia maritima*	400	Elevated expression of *NHX1*, *SOS1*, *HKT1*, *KT1,* and *VHA-E1* genes for ion homeostasis Improvement in SOD, CAT, and POD activities, and the levels of total phenolics and total flavonoids in leaves, for reduction of oxidative stress Enhanced levels of proline and soluble sugars in leaves for osmotic adjustment	[94]
**11**	*C. quinoa*	400	K^+^ retention in the cytosol through the unchanged or reduced expression of *CqGORK* and *CqSKOR* in the roots of salt-tolerant accessions	[95]
**12**	*Urochondra setulosa*	500	Improved biosynthesis of proline, glycine betaine, and trehalose for osmotic adjustment Upregulated transcript levels of L-ascorbate peroxidase, monodehydroascorbate reductase, CAT, and POD for an efficient antioxidant defense system Upregulated expression of gene encoding Na^+^/H^+^ exchanger for Na^+^ sequestration	[96]
**13**	*Cakile maritima*	400	Enhanced levels of amino acids like asparagine, alanine, lysine, proline, leucine, glycine, and threonine for amino acid biosynthesis Improved levels of sugars, such as glycerol, xylulose, and myo-inositol for osmoprotection Enhanced levels of gamma amino butyric acid (GABA) and glutamic acid for osmotic adjustment and photorespiration process	[97]
**14**	*Karelinia caspia*	200	Secretion of salt ions through salt glands *KcSOS1* indirectly modulated the K^+^ uptake/transport system for K^+^/Na^+^ ion homeostasis	[63]
**15**	*Suaeda salsa*	200	Endogenous levels of GA_3_, GA_4_, ABA, and BR were enhanced in stems at the end of the vegetative period Endogenous levels of GA_3_, GA_4_, IAA, and ZR were enhanced in flower organs during the flowering period Expression of genes related to GA_3,_ GA_4_, GR, IAA, BR, and ABA were upregulated in flowers	[98]
**16**	*Hordeum brevisubulatum*	750	The expression of *HbHAK1* confers salt tolerance by enhanced uptake of K^+^	[99]
**17**	*Zoysia macrostachya*	300	Secretion of salt ions through salt glands Higher levels of soluble sugars and proline for osmotic adjustment Higher activity of SOD, CAT, and POD protected the plant from salt-induced oxidative stress	[100]
**18**	*Sarcocornia fruticosa*	60	Dilutes the level of salt ions and stores water in plant tissues using succulence mechanism	[85,86,87]
	*Achras sapota*	120		
	*S. drummondii*	500		
**19**	*H. salicornicum*	400	Improved levels of proline, total free amino acids, and total soluble sugars for better osmotic adjustment Higher activity of SOD, as well as a higher level of GSH and the ratio of AsA/dehydroascorbate (DHA) and GSH/oxidized glutathione (GSSG), protected the plant from salt-induced oxidative stress	[101]
**20**	*A. tripolium*	300	Improved levels of glycine betaine and proline for osmotic adjustment Increased levels of total phenolics and flavonoids for reducing oxidative stress	[102]
**21**	*Leptochloa fusca*	450	Higher accumulations of proline and total soluble proteins for osmotic adjustment Improved activity of phenylalanine ammonia-lyase (PAL) and antioxidant enzymes SOD, CAT, and APX for mitigating oxidative damage The enhanced mRNA level of SOS1, PMA, and NHX1 in both roots and shoots	[103]
**22**	*C. maritima*	400	Elevated expression of *SOS1*, *SOS2,* and *SOS3* gene The activity of PMA and the SOS1 Na^+^/H^+^ antiporter helps to exclusion of Na^+^	[104]
**23**	*Eutrema parvula*	300	The transcript level of *EpHKT1;2* facilitated less Na^+^ and more K^+^ accumulation for maintaining improved K^+^/Na^+^ homeostasis	[105]
**24**	*Salvadora persica*	750	Increased accumulation of sugar and sugar alcohols like xylose, glucose, galactose, rhamnose, glycerol and myo-inositol help relieve salt stress-induced osmotic stress Enhanced levels of ABA and JA synergistically help in stomatal closure, water conservation and regulation of ion transport	[106]
**25**	*Pongamia pinnata*	500	Elevated expression of *SOS1*, *SOS2*, *SOS3*, *HKT1*, and ABA biosynthetic (*NCED*) and receptor (*PYL4*) genes in roots for ion homeostasis Enhanced *NHX1* expression in leaves for effective Na^+^ sequestration	[107]
**26**	*A. halimus*	171	Elevated expression of Na^+^/H^+^ antiporter gene in both roots and leaves for ion homeostasis Improved AsA concentration and CAT activity for ROS detoxification Improved contents of proline, soluble sugars and glycine betaine for osmotic adjustment	[108]
**27**	*Sesuvium portulacastrum*	120	Increased concentrations of proline, glycine betaine, and total soluble sugars for osmotic adjustment Enhanced activities of APX, GR, GPX, CAT, and SOD for detoxifying ROS Reduced the EC and salt adsorption ratio from 0.6 to 0.2 dS m^−1^ and 20.9 to 13.5 (mmol L^−1^)^0.5^, respectively	[109]
**28**	*Zygophyllum xanthoxylum*	50	The tonoplast NHXs facilitated efficient compartmentalization of Na^+^ into vacuoles *ZxAKT1* helped in uptake and long-distance transport of K^+^	[110]
**29**	*Puccinellia tenuiflora*	150	*PtNHX1* efficiently compartmentalized the excess Na^+^ into vacuoles of mesophyll cells *PtSOS1* enhanced Na^+^ loading into the xylem of roots *PtHKT1*; 5-induced loading of excess Na^+^ from xylem into xylem parenchyma cells	[111]
**30**	*Kosteletzkya virginica*	390	Upregulation of proline metabolism genes (*KvP5CS1* and *KvOAT*)	[112,113]

Abbreviations: ABA, abscisic acid; AKT, *Arabidopsis* K^+^ transporter; AsA, ascorbate; APX, ascorbate peroxidase; BR, brassinosteroids; CAT, catalase; CCC, cation-chloride cotransporter; CLC, chloride channel; EC, electrical conductivity; GA_3_/GA_4_, gibberellic acid; GPX, glutathione peroxidase; GORK, gated outwardly rectifying K^+^ channel; GR, glutathione reductase; GSH, glutathione; HKT, high affinity potassium transporter; HAK, high affinity potassium uptake transporter; IAA, indole acetic acid; JA, jasmonic acid; KT, potassium transporter; Kmt, Shaker type-potassium channel; lmt, myo-inositol O-methyl transferase; NADP, nicotinamide adenine dinucleotide phosphate; NCED, 9-cis-epoxycarotenoid dioxygenase; NHX, Na^+/^H^+^ exchanger; NRT, nitrate transporter; OAT, ornithine-δ-aminotransferase; PMA, plasma membrane H^+^-ATPase; POD, peroxidase; PYL, pyrabactin resistance 1-like; P5CS, delta 1-pyrroline-5-carboxylate synthase; ROS, reactive oxygen species; SKOR, stelar K^+^ outward rectifier; SOS, salt overly sensitive; SOD, superoxide dismutase; TCA, tricarboxylic acid; TPK, two pore K^+^ channel; VAP, vesicle-associated protein; VAMP, vesicle-associated membrane protein; VHA, vacuolar H^+^-ATPase; Vmac, V-ATPase subunit c; VP, vacuolar H(^+^)-pyrophosphatase; ZR, zeatin riboside.

## 6. Biochemical Mechanisms Associated with Halophyte Adaptation to Salinity

Plants, including halophytes, adopt several biochemical strategies to adapt against salinity-caused adverse effects. For instance, ion homeostasis, osmotic adjustment, and ROS-detoxification are among the important biochemical mechanisms used by plants to protect their cellular constituents from the detrimental effects of excessive salt [76] (Figure 2 and Figure 3; Table 1). Here, we summarize these three biochemical mechanisms associated with halophyte responses and tolerance to salt stress.

### 6.1. Ion Homeostasis

Various channels and transporters are actively involved in accommodating Na^+^, Cl^−^, and K^+^ in the tissues of halophytic plants. It is well known that the uptake of Na^+^ from soils to roots occurs mainly via the apoplastic and symplastic pathways [34]. To transport salt ions via the symplastic pathway, transporters and channels are needed. Generally, non-selective cation channels (NSCCs), including cyclic nucleotide-gated channels (CNGCs) and glutamate receptor-like channels (GLRs), high-affinity K^+^ transporters (e.g., HKT2 and HAK5) and aquaporins, such as plasma membrane intrinsic protein (PIP) isoforms, are mainly responsible for root-Na^+^ uptake (Figure 2). The influx of Na^+^ can also be aided by low-affinity cation transporter (LCT1) and *Arabidopsis* K^+^ transporter 1 (AKT1) (Figure 2) [34]. The inward movement of Cl^−^ in root cells is mediated by H^+^/Cl^−^ symporters, Cl^−^/H^+^ co-transporters, and nitrate transporters (NRTs). During high salinity, several anion channels, including slow anion channels (SLAC channels), are involved in the passive transport of Cl^−^ [114]. Cation chloride cotransporters (CCCs) help in Cl^−^ influx in the xylem and shoot tissues [115]. The chloride channels (CLCs) found on tonoplast are responsible for Cl^−^ sequestration [75] (Figure 3). In the presence of Na^+^, K^+^ is competitively uptaken via HKT2, HAK5, and AKT1 [11].

To prevent the accumulation of excess salt in their cytosol, halophytes mainly adopt two strategies: (i) prevention of salt entrance to the roots and (ii) vacuolar sequestration of salt ions [76] (Figure 3). However, these mechanisms are not mutually exclusive, and a particular plant may employ multiple strategies depending on the circumstances. For example, ion homeostasis in the cytosol is a unique strategy that plants follow to limit salt ion accumulation by compartmentalizing them effectively, which is essential for plant survival under salinity-triggered hostile conditions. The SOS system is crucial for perceiving salt-mediated Ca^2+^ signals, as well as for the exclusion of excess Na^+^ by roots, loading of Na^+^ into the xylem and sequestration of excess Na^+^ into the vacuoles using Na^+^/H^+^ exchangers (NHXs) [11,116] (Figure 3). The SOS system is predominantly comprised of SOS3/SOS3-like calcium-binding proteins (SCaBPs), SOS2 (a serine/threonine-protein kinase) and SOS1 (a plasma membrane Na^+^/H^+^ antiporter) [11,76] (Figure 3). Importantly, SOS4 (pyridoxal kinase) and SOS5 have recently been discovered, with SOS4 being involved in the biosynthesis of pyridoxal-5-phosphate (vitamin B6) that plays a key role in Na^+^/K^+^ homeostasis in plants, and SOS5 being a putative cell surface adhesion protein that is required for normal cell expansion [116]. Generally, SOS3 relays salt-triggered Ca^2+^ signals to SOS2 kinase, forming an SOS3–SOS2 protein complex that phosphorylates SOS1 to induce Na^+^ efflux [11,84] (Figure 3). The elevated expression of the *SOS1* gene and its correlation with the improvement of salt tolerance have been reported in the obligate halophyte *Cakile maritima* [104], ornamental halophyte *Lobularia maritima* [94], and recretohalophyte *Karelinia caspia* [63] (Table 1). Likewise, Marriboina et al. [107] and Arbelet-Bonnin et al. [104] reported that the simultaneous expression of *SOS1, SOS2,* and *SOS3* contributed to high salinity tolerance in the phreatophyte *Pongamia pinnata* and obligate halophyte *C. maritima* species, respectively (Table 1).

NHXs are also helpful in controlling vacuolar Na^+^ influx and maintaining optimum intracellular Na^+^/K^+^ homeostasis [76] (Figure 3; Table 1). For instance, under saline conditions, the perennial woody halophyte *Nitraria sibirica* could maintain an optimal K^+^/Na^+^ balance at the tissue- and cell-specific levels via vacuolar Na^+^ sequestration by boosting the activities of both vacuolar H^+^-ATPase and H^+^-PPase, and upregulating the expression of *NsVHA*, *NsVP1,* and *NsNHX1* [93]. Meanwhile, *N. sibirica* increased the expression of *two-pore K^+^* (*NsTPK*) at the transcriptional level, promoting K^+^ efflux from the vacuoles to the cytoplasm for cytosolic K^+^ homeostasis [93] (Table 1). In addition, Tran et al. [75] found that salt-exposed *M. crystallinum* enhanced the expression of genes encoding tonoplast antiporters H^+^/Cl^−^ (i.e., *McCLC1*), Na^+^/H^+^ exchangers (i.e., *McNHX1*), and V-ATPase subunit c (i.e., *McVmac1*) for sequestration of Cl^−^ and Na^+^ into the vacuoles for conferring salt tolerance in *M. crystallinum* (Table 1).

### 6.2. Osmotic Adjustment

Plants can attain osmotic adjustment by employing two strategies: (i) increased uptake of inorganic ions, and (ii) *de novo* synthesis of organic osmolytes to maintain turgor pressure and organellar volume within the growing plant cells [11]. Fundamentally, Na^+^ and Cl^−^ in vacuoles, as well as K^+^ and compatible organic solutes like proline and sugars in the cytoplasm, are crucial to assure osmotic adjustment in the cells in a salt-caused osmotic environment (Figure 3) [117]. Generally, salinity-induced K^+^ efflux from the cell takes place via two major types of channels, including guard cell outward rectifying K^+^ (GORK) channels and/or ROS-activated NSCCs [11] (Figure 2). Although GORK channels play a central role in stress-induced K^+^ loss from the cytosol, comprehensive research into the molecular background of GORK channel-mediated osmotic balance in halophytes is still elusive. Nonetheless, by studying two contrasting *C. quina* accessions, Kiani-Pouya et al. [95] have recently revealed that the transcript levels of *CqGORK* in roots remained unchanged or decreased in salt-tolerant accession but increased in salt-sensitive accession, implying GORK’s role in K^+^ retention for better osmotic adjustment in the cells of salt-sensitive *C. quina*. Furthermore, the ion selectivity approach of HAKs/HKTs has been reported to confer greater uptake of K^+^ in *Eutrema parvula* [105] and *Hordeum brevisubulatum* [99] and, thus, contributed to improved K^+^/Na^+^ homeostasis (Table 1). Nonetheless, future studies should be taken to decipher how the movement of K^+^ between intra-cellular and inter-cellular membranes by K^+^ transporters contribute to the adjustment of the osmotic environment in salt-exposed halophytic plants.

Halophytes often produce a multitude of osmoprotectant molecules, including amino acids, soluble sugars, sugar alcohols and glycine betaine to maintain osmotic balance under salinity [118] (Figure 3; Table 1). The accumulation of proline and the elevated expression of *delta 1-pyrroline-5-carboxylate synthase* (*P5CS*) gene involved in proline biosynthesis, were associated with improved salt tolerance in *M. crystallinum* [75] and *Urochondra setulosa* [96] (Table 1). In addition, salt tolerance by glycine betaine has also been recorded in *Atriplex halimus* [108] and *Sesuvium portulacastrum* [109] (Table 1). It is worth mentioning that metabolite profiling in halophytes has recently gained momentum as a means of gaining a comprehensive understanding of the levels of accumulated osmolytes in relation to salt tolerance. For instance, Panda et al. [88] and Kumari and Parida [106] uncovered the salt-responsive metabolites and metabolic pathways in xero-halophyte *Haloxylon salicornicum* and facultative halophyte *Salvadora persica* plant, respectively (Table 1). They found that salt-exposed *H. salicornicum* and *S. persica* had greater levels of sugars, sugar alcohols, and amino acids, indicating that these osmolytes might contribute to the osmotic adjustment, resulting in salt tolerance in these plants (Table 1). Furthermore, metabolite profiling of *C. maritima* exposed to 400 mM sodium chloride (NaCl) showed that proline, glycine betaine and gamma amino butyric acid (GABA) significantly accumulated, while sugar levels surprisingly decreased, indicating that amino acids and amino acids-derived molecules played the primary roles in osmoprotection to cope with salt stress in this halophyte [97] (Table 1). Nonetheless, further exploration of metabolite changes in other halophytes should be undertaken to deepen our existing knowledge on putative salt tolerance mechanisms in halophytes.

### 6.3. ROS-Detoxification

Plants rely on efficient antioxidant defense mechanisms to protect them from oxidative stress, induced by the overaccumulation of ROS under elevated salinity [119]. To alleviate ROS-induced damage, plants possess a robust antioxidant defense system, which comprises of non-enzymatic antioxidants like flavonoids, AsA, GSH, and tocopherols, and enzymatic antioxidants, such as SOD, CAT, APX, glutathione reductase (GR), monodehydroascorbate reductase (MDHAR), dehydroascorbate reductase (DHAR), GPX, and GST [39] (Figure 3). The improved salt tolerance of various halophytes is associated with their ability to maintain ROS homeostasis by a proper activation of their antioxidant system under salt stress [89,90,94,100,101,120]. For example, in *H. salicornicum* plant, the higher activity of SOD, higher level of GSH and higher ratios of AsA/dehydroascorbate (DHA) and GSH/oxidized glutathione (GSSG) play a pivotal role in scavenging of ROS to withstand oxidative stress and improve salt tolerance [101] (Table 1).

In response to salt stress, halophytic plants modulate the levels of endogenous phytohormones in order to regulate various defense mechanisms, including antioxidant defense [91,121]. Detailed roles of phytohormones in halophyte responses to abiotic stresses have been reviewed in-depth elsewhere [121]. In halophytes, phytohormone accumulation varies depending on the types of halophytes exposed to salt stress. For instance, jasmonic acid (JA) content increased in *S. persica* [106] but dropped in *Aster tripolium* [91] and *H. salicornicum* [88] under salt stress (Figure 3, Table 1). The levels of salicylic acid did not alter significantly in *S. persica* [106] and *H. salicornicum* [88], while being remarkably dropped in *A. tripolium* [91] (Figure 3). The content of gibberellic acid (GA) increased noticeably in *S. salsa* [98], while it decreased in *H. salicornicum* [88] (Figure 3, Table 1). Because JA can be produced via linolenic acid oxidation [122], the JA level in plants is linked to the amount of ROS and the rate of ROS-induced lipid peroxidation [123]. Indeed, a lower level of JA was positively correlated with the reduced level of ROS under salt stress [88]. On the other hand, the level of abscisic acid (ABA), which acts as a critical hormone in stomatal closure to reduce transpiration, significantly increased in *Lycium humile*, *S. persica*, *H. salicornicum* and *A. tripolium* in response to salt stress [88,91,92,106] (Table 1). Panda et al. [88] reported that enhanced ABA accumulation helped in reduction of oxidative stress in salt-exposed *H*. *salicornicum*. The generation of ROS and subsequent redox processing are an integral part of hormone regulation and have a role in controlling plant development and stress tolerance [124]. Therefore, future studies should focus on spatial-temporal regulation of ROS generation and identification of proteins that could detect changes in ROS. Such information may help us further understand the crosstalk among multiple hormones signaling pathways in halophytes, and their hormone-modulated stress-tolerant phenotype.

## 7. Halophyte-Mediated Phytoremediation of Salinity-Affected Soils

Extensively used methods to reclaim salt-affected areas are leaching and adding chemicals and organic amendments [125]. Leaching facilitates the movement of salt particles from the surface horizon to the deeper horizon. However, lack of water during leaching and its effects on reduced soil stability, contents of total nitrogen and total organic carbon, microbial functions and overall soil fertility limit its practice in salt-contaminated areas [126]. Chemical approach involves an ion exchange process through existing and/or adding calcium carbonate (CaCO_3_) or other chemical substitutes to the soils; however, poor solubility and quality and high prices restrain further expansion of chemical uses in reclaiming of saline soils [126,127]. Interestingly, optimal rates of the different organic amendment (not more than 50 T ha^−1^) can efficiently increase dissolution of CaCO_3_ [128], as well as improve soil-physical (e.g., structure, permeability, water holding capacity, etc.) and chemical properties (e.g., pH, cation exchange capacity, etc.) that favors plant growth and microbial activities [129]. However, benefits are restrained due to the risk of releasing various chemicals, including heavy metals, phenolic compounds, ethylene and ammonia, excess salts, and organic acids from uncategorized organic wastage [129,130,131,132]. All of these constraints have forced the farmers to look for alternative options that are environment-friendly, cost-effective, and aesthetically pleasing.

Phytoremediation, popularly known as vegetative bioremediation or biological reclamation, is a promising option over other remediation techniques, which involves cultivating certain salt-tolerant species to remove salt ions from the soil to restore salinity-degraded soils [20,126,133]. Its acceptance is increasing progressively due to numerous benefits. For example, phytoremediation (i) is economically viable, as it does not require the purchase of chemicals; (ii) enhances nutrient availability for plants in soil, which helps boost crop yield; (iii) removes excess salts more uniformly, even from higher depth when compared with gypsum; and (iv) is environment-friendly as it sequestrates carbon in both above- and below-ground biomasses [47]. Unfortunately, information regarding mechanistic insights into phytoremediation of saline soils using halophytes is still elusive. Because of complicated interactions, it is challenging to visualize plants’ complete and holistic mode-of-action in the saline soil remediation process, and their beneficial effects on soil properties. According to our current knowledge, we have illustrated a well-established and recognized underlying phytoremediation mechanism of halophytes in Figure 4.

First, halophytic plants can help remove salts by taking salt ions through root system and store them in aboveground plant parts (Figure 4). *S. salsa* has recently been successfully used to reclaim saline soil in the northwest of China due to its salt extraction potential, which ranges from 3.75 to 3.91 T ha^−1^ year^−1^ [133]. It is important to note that *S. salsa* can theoretically remove 3.0–3.8 T Na^+^ ha^−1^ with a population density of 15 plants m^−2^ [134]. Liang and Shi [134] also reported that the aboveground biomass of *S. salsa* plants accumulated approximately 27 mg of salt g^−1^ dry weight. Shaygan et al. [135] demonstrated that the shoots of *Tecticornia pergranulata* could accumulate 98 g of salt Kg^−1^ dry weight. Sodium chloride is often used to lower the freezing point of water on North American highways to prevent the risk of vehicle crashes and injuries [136]. The same study [136] showed that *A. patula* would take six years to remove all the salt from an area contaminated with 1540 g Cl^−^ per m^2^, if salt application was halted. On the other hand, since salt is applied every winter, *A. patula* would be a potential candidate for the management of salt amount accumulated over time in the roadside soil and groundwater.

Second, some studies suggest that planting halophytes could reduce salt ion accumulation in soils by improving soil physical properties. Briefly, the deep-rooted halophytic plants have the potential to act as tillage tools. This is popularly referred to as biological drilling and has been reported to stabilize soil structure by reducing soil bulk density, thereby increasing soil porosity and hydraulic properties that augment Na^+^ leaching and replace it with other cations, besides inducing higher nutrient use efficiency [137,138] (Figure 4). In line with this finding, planting *S. salsa* with cotton (*Gossypium hirsutum*) as an intercropping system significantly decreased soil salinity and bulk density, while increasing soil porosity, soil organic carbon, root growth, total aboveground biomass, and cotton yield, when compared with the traditional monocropping system [134]. In addition, *A.*
*nummularia* showed improvement of hydraulic conductivity and reduction in bulk density, which led to enhanced macropores and water infiltration rate under 40.86 dS m^−1^ salinity level [138]. Ashraf et al. [139] also reported that planting halophytes reduced bulk density, while it increased soil porosity and hydraulic conductivity, which eventually improved salt ion leaching from saline-alkali soil.

Third, root respiration and decomposition of organic matter play a pivotal role in reducing Na^+^ content in the saline-sodic soils because they can increase CO_2_ concentration in the soil atmosphere [140]. In brief, irrigation water and soil moisture-induced availability of soil water react with CO_2_ to form carbonic acid (H_2_CO_3_) (Figure 4). The H_2_CO_3_ dissociation and N_2_-fixation in halophyte roots lead to the release of the proton (H^+^), which causes dissolution of calcite (CaCO_3_) to produce Ca^2+^, H_2_O and CO_2_ [140] (Figure 4). The released Ca^2+^ in the soil could facilitate the removal of Na^+^ from the cation exchange sites of soil colloid [140].The exchanged Na^+^ content in soils is then eliminated via two routes: (i) uptake through halophytic plant roots, and (ii) leaching as a result of flooding irrigation prior to sowing, which ultimately contributes to the reduction in soil sodicity and salinity (Figure 4).

**Figure 4 ijms-22-10733-f004:**
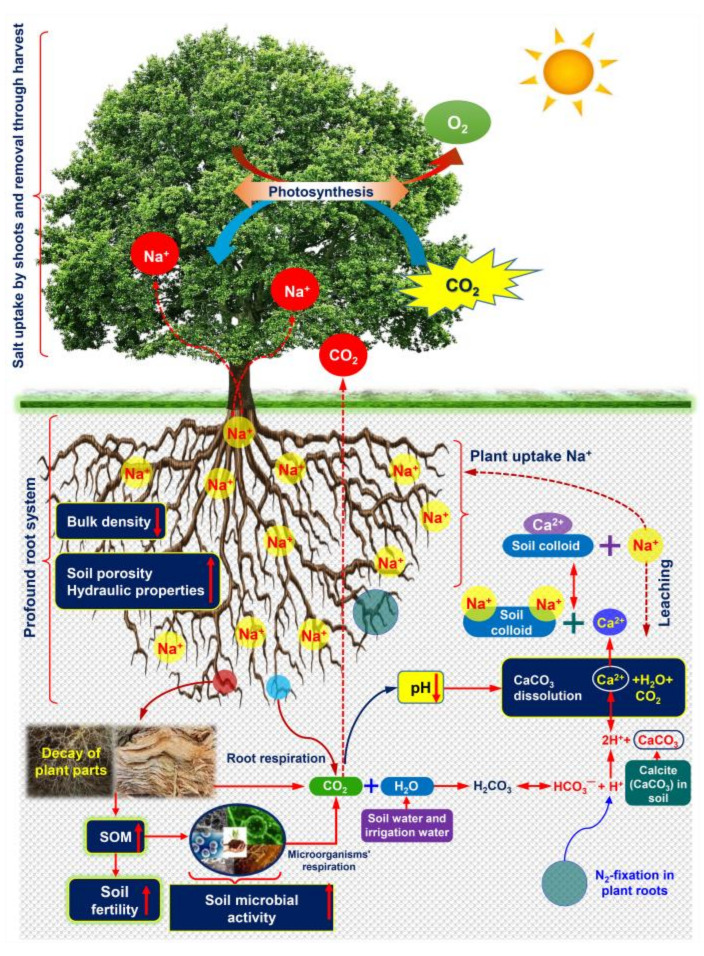
Putative phytoremediation process by halophyte plants in salinity-affected areas, and their beneficial effects on soil properties (modified from Qadir et al. [140] and Jesus et al. [126]. Deep-rooted plants help reduce soil bulk density and increase soil porosity, enabling storage of excessive salt ions in deeper soil layers through leaching. On the other hand, the reclamation of saline soils needs a source of Ca^2+^ that replaces excessive Na^+^ from the cation exchange sites of soil colloid. Decay of plant parts enhances soil organic matter (SOM) content, leading to the improvement of soil fertility. The microbes-mediated SOM decomposition and root respiration increase the concentration of carbon dioxide (CO_2_) in the soil atmosphere. Water (H_2_O) from irrigation and soil moisture sources reacts with CO_2_ to generate carbonic acid (H_2_CO_3_). In halophyte roots, H_2_CO_3_ dissociation and N_2_-fixation release proton (H^+^) that enhances dissolution of calcium carbonate (CaCO_3_) to release Ca^2+^, H_2_O and CO_2_. The released Ca^2+^ could help remove Na^+^ from cation exchange sites of soil colloid. Finally, the exchanged Na^+^ can be uptaken by roots, and compartmentalized in aboveground shoots or leached into the deeper layer of soils via irrigation. In non-calcareous soils, an increase in CO_2_ induces the release of H^+^, which results in a decrease in the pH of the soil. The acidic nature of soil enhances the dissolution of CaCO_3_, which helps reduce the amount of salt ions in soils via Ca^2+^ exchange with Na^+^.

Nonetheless, in non-calcareous soils, augmentation of CO_2_ results in the lowering of soil pH that enhances dissolution of CaCO_3_ to make Ca^2+^ available to interchange with Na^+^ [140]. Planting two facultative halophytes, *S. verrucosum* and *Bacopa monnieri*, for 240 days reduced the electrical conductivity (EC) value from 11.13 to 7.97 dS m^−1^, pH from 7.84 to 7.42, while improving soil porosity from 54.71 to 57.23%. Notably, the combined plantation of these two halophytes had a phytodesalination capacity of 1.21 T Na^+^ ha^−1^, which helped prepare the soil for *Zea mays* to thrive and boosted yields from 7.2 to 8.5 T ha^−1^ [141]. *S. monoica*, a salt marsh halophyte, has been shown to lower the soil EC value from 4.75 to 2.10 dS m^−1^ and pH from 8.40 to 6.73 [142].

However, the use of halophytes as a desalinization tool cannot be guaranteed in all situations, since this approach is affected by environmental inconsistencies, such as rainfall and temperature, or cultivation modes, such as irrigation and non-irrigation, as well as the risk of recycling back of the salt particles through plant parts [143]. Therefore, robust assessments of the potential of various halophytes by analyzing their mode-of-action are needed prior to their application in the desalinization of salt-affected areas. In addition, further research will be required to testify the suitability of the plant-uptake-mediated phytoremediation process, and if found significant, it will open a new avenue for restoring salinity-affected areas, notably non-calcareous soils and soils under non-leaching circumstances.

## 8. Potential of Halophytes as Genetic Resources for the Engineering of Crops with Improved Salt Tolerance

Halophytes can survive better under ambient salinity than glycophytes, which might be attributable to proper modulation of genes that play crucial roles in triggering their salt tolerance mechanisms [76,144]. High-throughput transcriptome sequencing is increasingly being used to understand the responses of plants to abiotic stresses at gene expression level [145]. Recently, Guo et al. [146] conducted a comparative transcriptome analysis of the model halophyte *Puccinellia tenuiflora* and several glycophytic Gramineae plants to uncover their underlying molecular responses to excessive salt stress. They found that the greater expression of the gene families involved in K^+^ uptake, sucrose biosynthesis, pentose phosphate pathway and flavonoid biosynthesis might be associated with better growth of *P. tenuiflora* under high salinity conditions. In *Salicornia persica*, Aliakbari et al. [147] identified 1595 differentially expressed genes (DEGs) between the control and salt-treated plant shoots, including those encoding transcription factors, transporters, and protein kinases. Functional annotation analyses indicated that improved energy homeostasis and the synthesis of primary metabolites might play pivotal roles in salt tolerance of *S. persica*. Gene network analysis revealed that Ca^+^ signaling, ABA signaling, and Na^+^ compartmentalization are essential components in conferring salt tolerance in *S. persica*. Han et al. [148] identified 399 upregulated and 327 downregulated genes in *S. rigida* plants in responses to different regimes of soil salinity (100, 300, and 500 mM NaCl). Gene ontology (GO) enrichment analysis suggested that the downregulated genes were related to organic substance transport, nitrogen compound transport and intracellular protein transport. In contrast, the upregulated genes were involved in cell wall biogenesis, such as cell wall assembly, extracellular matrix organization, and cell wall formation [148].

Moreover, RNA-seq identified 9144 DEGs in *A. centralasiatica* leaves under salt stress, with 3819 upregulated and 5325 downregulated genes [81]. GO enrichment analysis showed that salt stress dramatically increased the expression of genes associated with ion transport, ABA-dependent signaling pathway, ROS-scavenging and transcriptional regulation (HB-7 and MYB78) pathway [81]. A *de novo* transcriptome profiling in the halophyte *U. setulosa* revealed the upregulation of genes encoding photosynthetic enzymes, transcription factors, mitogen-activated protein kinases (MAPKs), transporter proteins, cell membrane proteins, antioxidant enzymes, and enzymes for synthesis of compatible solutes by salinity [96]. Wang et al. [100] revealed many DEGs, specifically 4903 upregulated and 3800 downregulated genes in *Z. macrostachya,* in response to 300 mM NaCl. Such expression change in a large number of genes may contribute to improved plant growth performance by modulating plant hormone signal transduction, ion homeostasis via salt secretion, and osmoregulatory substance accumulation and prevention of oxidative damage through the activation of an efficient ROS-scavenging system.

Because they are genetically equipped with salt tolerance potential, halophytes could be a vital genetic resource for developing transgenic plants with better adaptability to salt stress. In fact, many transgenic plants have been developed via the introduction of genes from halophytic plants to enhance salt tolerance. Most of these genes encode antiporters, ion transporters, ROS scavengers, antioxidants, and proteins that play crucial roles in plants′ signal transduction and functional stabilization [149,150]. For example, transgenic poplar plant ectopically expressing a *N. sibirica* vacuolar Na^+^/H^+^ antiporter gene (*NsNHX1*) displayed a more remarkable improvement in salt tolerance through improved compartmentalization of Na^+^, more efficient photosynthesis, greater activity of antioxidant enzymes, and enhanced osmotic adjustment [151] (Table 2). Ectopic overexpression of a high-affinity K^+^ transporter-encoding gene, *SbHKT1*, from *S. bigelovii* in cotton improved salt tolerance of transgenic plants by enhancing K^+^ uptake capacity, K^+^/Na^+^ homeostasis, and the scavenging of ROS through increased activities of antioxidant enzymes, including SOD, POD, and CAT [152]. On the other hand, transgenic *Arabidopsis* ectopically expressing *Zygophyllum xanthoxylum* nitrate transporter 1-encoding gene *ZxNRT1.5* displayed improvement in NO_3_^−^ uptake and root-to-shoot NO_3_^−^ translocation, which eventually led to increased biosynthesis of amino acids in the transgenic plants that helped in an osmotic adjustment under salt stress [153,154]. Further studies are required to uncover the comprehensive molecular mechanisms underlying *ZxNRT1.5* involvement in regulating Na^+^, K^+^, and Cl^−^ transport. In addition, ectopic overexpression of the *N. tangutorum calcineurin B-like protein-interacting protein kinase 11* (*NtCIPK11*) gene from *N. tangutorum* in *Arabidopsis* improved seed germination and root length, as well as altered the transcription of genes encoding key enzymes involved in proline metabolism and K^+^ transportation in transgenic plants subjected to salt stress [155,156] (Table 2). Transgenic tobacco plants ectopically expressing several novel genes from *S. brachiata**,* such as *S. brachiata stress-related protein* (*SbSRP*), *S. brachiata salt inducible-1* (*SbSI-1*), *S. brachiata myeloblastoma15* (*SbMYB15*), *S. brachiata dehydration-responsive element-binding 2A* (*SbDREB2A*), and *S. brachiata RNA poly III complex 5-like subunit* (*SbRPC5L*), and *Aeluropus lagopoides,* such as *AlRab7* (*AlRabring7*), displayed higher accumulations of phytohormones (e.g., indole acetic acid) and compatible solutes (e.g., proline), improved gas exchange features, reduced membrane damage, improved antioxidant capacity, lower accumulations of toxic ions, ultimately resulting in better biomass and growth performance of transgenic plants under salt stress [157,158,159,160,161,162] (Table 2). In addition, ectopic expression of *L. maritima*
*vacuolar H+-ATPase subunit E1 (LmVHA-E1*) from *L. maritima*, *Tamarix hispida dehydration-responsive element-binding* (*ThDREB*) from *T. hispida*, *PutNHX1* or *PutAPX* from *P. tenuiflora**,* or *SeNHX1* from *S. europaea* in *Arabidopsis* plants improved ROS-scavenging, as well as compartmentalization of toxic Na^+^ into vacuoles to enhance salt tolerance of transgenic plants [150,163,164] (Table 2). Furthermore, transgenic tobacco harboring *Iris lactea* tonoplast Na^+^/H^+^ antiporter gene *IlNHX* showed elevation in the compartmentalization of Na^+^ into vacuoles via enhanced vacuolar V-ATPase activity, which contributed to the maintenance of K^+^/Na^+^ homeostasis [165] (Table 2).

Halophytes with intrinsic and efficient salt tolerance abilities can be used as models for improving crop and/or plant responses to salt stress (Table 2). Furthermore, because crops are predominantly glycophytes, it is critical to highlight the molecular differences between these two plant groups. Such findings could be the key to guiding and optimizing glycophyte transformation using halophytic candidate genes, thereby providing a new avenue for study aimed at preparing crops to cope with elevated soil salinity. However, despite their impressive implication, information regarding identification, metabolic functions, and isolation of crucial genes from halophytes that are associated with salt tolerance is still limited. Therefore, identification and characterization of the novel salt tolerance-associated genes from halophytes, and their subsequent utilization in genetic engineering are important areas of plant research, which deserves more attention to sustainably increase crop production for achieving global food security under the era of climate change.

## 9. Conclusions and Future Perspectives

Salt-induced land degradation restrains crop production and, thus, endangers sustainable food production and food security in the world. Climate-driven changes and anthropogenic activities have aggravated salinity problems, pressing the need to cultivate plant species that can withstand ambient salinity. Halophytes are a diverse group of plants with excellent adaptabilities to high salinity at both cell and whole-plant levels, and thus, have significant potential to meet this such cultivation challenge and be successfully integrated into future farming systems. Several halophytes are able to grow naturally in ambient salinity through the adoption of intrinsic salt tolerance mechanisms. These halophytes could be used as potent commercial alternatives to conventional crops, although commercialization of halophytic products has just started. At the same time, the cultivation of halophytic species in salinity-affected areas can reduce salt contamination of soils. Thus, this phytoremediation technique could be the most potent and viable strategy to remediate salinity-affected lands and facilitate crop production in salt-prone areas.

Despite its tremendous potential in soil salinity management, phytoremediation has limitations since it is a time-consuming process and requires screening out of suitable species. Therefore, future research should focus on sorting out halophytic plants with multiple salt tolerance mechanisms, which can produce higher biomass for removing the maximum quantity of salts from contaminated soils. Furthermore, despite tremendous research advances on halophytes (for more than 100 years), some basic physiological, biochemical and molecular mechanisms are still unclear. Hence, in-depth research is eminently needed to unravel halophytes’ underlying salt tolerance mechanisms for better soil conservation. Furthermore, recent genetic and omic studies showed that salt tolerance is a complex trait that depends on whole plant responses rather than the action of a single gene or protein. Therefore, exploration of the salt tolerance potential of various halophytes and the development of transgenic plants with superior remediation features are crucial for sustainable agriculture in salinity-affected areas to reduce the curse of global climate change.

## Figures and Tables

**Figure 1 ijms-22-10733-f001:**
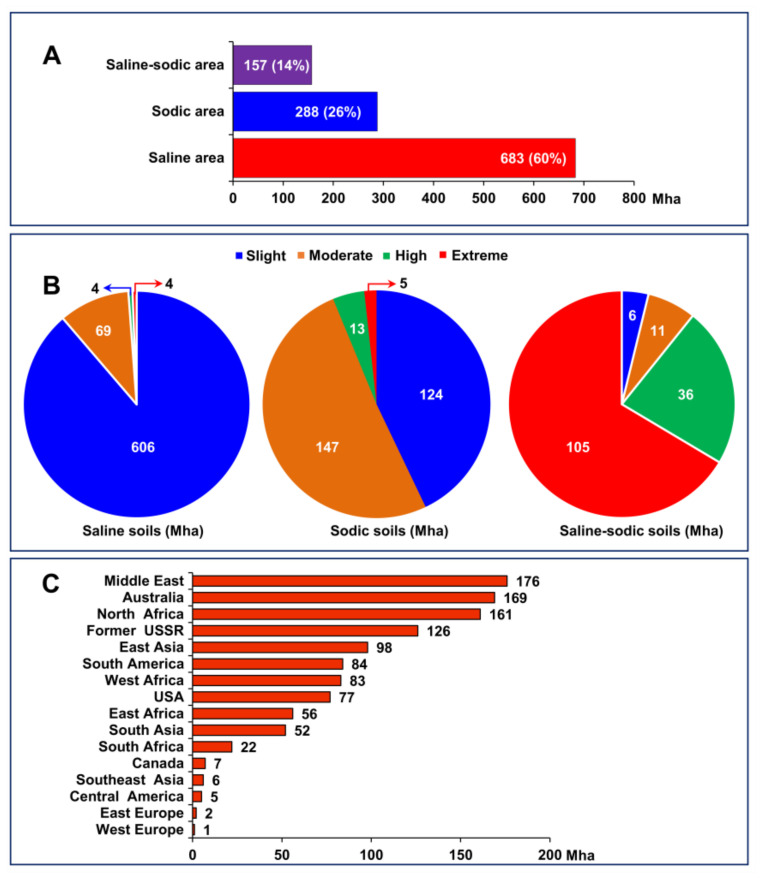
Overview on soil salinization across the world. (**A**) Salinity-affected soil categories *. (**B**) Types and severity levels of salinity-affected soils. (**C**) Salinity-affected soils in different parts of the world *. * Due to rounding, the sum may not resemble the actual statistics of salinity-affected lands. Mha, million hectares.

**Table 2 ijms-22-10733-t002:** Improvement of salt tolerance in plants by overexpressing various genes isolated from different halophytes.

Sl. No.	Halophytic Plants	Isolated Genes	Over-Expressed In	Improved Features Contributed to Salt Tolerance	References
**1**	*Nitraria tangutorum*	*NtCIPK11*	*Arabidopsis thaliana*	Seed germination rate, root length, plant growth and expression of proline biosynthesis-associated gene (e.g., *P5CS1*, *P5CS2*, *P5CR,* and *ProDH1*)	[156]
**2**	*Aeluropus lagopoides*	*AlRab7*	*Nicotiana tabacum*	Seed germination rate, phytohormone (indole acetic acid) accumulation, ion homeostasis status, gas-exchange features and antioxidant enzyme activities	[162]
**3**	*N. sibirica*	*NsNHX1*	*Populus alba*	Root system, chlorophyll content, RWC, proline level, Na^+^ compartmentalization and activities of antioxidant enzymes	[151]
**4**	*Salicornia bigelovii*	*SbHKT1*	*Gossypium hirsutum*	Seed germination rate, biomass accumulation, optimal K^+^/Na^+^ ratio and activities of antioxidant enzymes	[152]
**5**	*N. tangutorum*	*NtCIPK11*	*A. thaliana*	Seed germination rate, root length, and expression of K^+^ transporter gene (*AtHKT1*)	[155]
**6**	*Zygophyllum xanthoxylum*	*ZxNRT1.5*	*A. thaliana*	Transportation and uptake of K^+^ from roots to shoots	[154]
**7**	*Iris lactea*	*IlNHX*	*N. tabacum*	Higher vacuolar H^+^-ATPase (V-ATPase) activity for compartmentalization of Na^+^ into the vacuoles	[165]
**8**	*S. brachiata*	*SbRPC5L*	*N. tabacum*	Photosynthetic rate, ion homeostasis, membrane stability and overexpression of antioxidant-related genes (e.g., *NtAPX*, *NtPOX,* and *NtSOD*)	[161]
**9**	*Porteresia coarctata*	*PcINO1* and *PcIMT1*	*Oryza sativa*	Seed germination rate, growth and biomass, photosynthetic activity and inositol level	[166]
**10**	*Salix matsudana*	*SmCP*	*A. thaliana*	Seed germination rate, photosynthetic pigment levels, antioxidant enzyme activities and root’s ion homeostasis	[167]
**11**	*Eutrema halophilum*	*EhEm1*	*O. sativa*	Germination rate, levels of chlorophyll and proline, and activities of POD and lactate dehydrogenase	[149]
**12**	*S. brachiata*	*SbSRP*	*N. tabacum*	RWC, photosynthetic rate, and accumulations of compatible solutes (e.g., proline, sugars, free amino acids, starch, and polyphenols)	[157]
**13**	*Lobularia* *maritima*	*LmVHA-E1*	*A. thaliana*	Expression of stress-related genes (e.g., *AtNHX, AtP5CS, AtCAT, AtSOD, AtPOD,* and *AtLEA*)	[163]
**14**	*Tamarix hispida*	*ThDREB*	*N. tabacum*	Germination rate, root length, plant biomass, chlorophyll content, and activities of SOD and POD	[164]
**15**	*S. brachiata*	*SbSI-1*	*N. tabacum*	RWC, photosynthetic rate, membrane stability, K^+^/Na^+^ ratio, polyphenol content, and activities of antioxidant enzymes (e.g., SOD, CAT, and APX)	[158]
**16**	*Puccinellia tenuiflora* and *S*. *europaea*	*PutNHX1* and *SeNHX1*	*A. thaliana*	Na^+^ sequestration in the vacuoles, and K^+^ retention in the cytosol and vacuoles of root cells	[150]
**17**	*Reaumuria trigyna*	*RtNHX1*	*A. thaliana*	Seed germination rate, plant biomass, root length, photosynthetic pigments, RWC, K^+^/Na^+^ ratio, proline level, and activities of antioxidant enzymes (e.g., POD and CAT)	[168]
**18**	*Suaeda salsa*	*SsGPAT*	*A. thaliana*	Improved content of unsaturated fatty acid helps to alleviate the photoinhibition of photosystem (PSI)-I and PS-II	[169]
**19**	*Sesuvium portulacastrum*	*SpAQP1*	*N. tabacum*	Seed germination rate, root length, and activities antioxidant enzymes (e.g., SOD, POD, and CAT)	[170]
**20**	*Zoysia matrella*	*ZmVP1*	*A. thaliana*	Sequestration of Na^+^ into vacuoles, K^+^ assimilation, and activities of antioxidant enzymes (e.g., SOD, POD, and APX)	[171]
**21**	*S. brachiata*	*SbMYB15*	*N. tabacum*	Membrane stability, stomatal conductance, water use efficiency, photosynthetic rate, K^+^/Na^+^ ratio, and expression of stress-responsive genes (e.g., *LEA5*, *ERD10*, *LTP1*, *HSF2*, *ADC1*, *PLC3,* and *P5CS*)	[159]

**Abbreviations**: APX, ascorbate peroxidase; ADC, arginine decarboxylase; AQP, aquaporin; CIPK, CBL-interacting protein kinase; CAT, catalase; CP, cysteine protease; DREB, dehydration-responsive element-binding; ERD, early responsive to dehydration; EM, embryogenesis abundant; GPAT, glycerol-3-phosphate acyltransferase; HSF, heat shock transcription factor; HKT, high affinity potassium transporter; IMT, inositol methyl transferase; INO1, L-myo-inositol 1-P synthase; LTP, lipid transfer protein; LEA, late embryogenesis abundant; MYB, myeloblastoma; NRT, nitrate transporter; NHX, Na^+^/H^+^ exchanger; P5CS, delta 1-pyrroline-5-carboxylate synthase; P5CR, pyrroline-5- carboxylate reductase; ProDH, proline dehydrogenase, POX/POD, peroxidase; PLC, phospholipase c-like; P5CS, delta 1-pyrroline-5-carboxylate synthase; RWC, relative water content; RPC5L, RNA poly III complex 5 like; SOD, superoxide dismutase; SRP, stress-related protein; SI, salt inducible; VHA, vacuolar H^+^-ATPase; VHA-E1, vacuolar H+-ATPase subunit E1; VP, vacuolar H (^+^)-pyrophosphatase.

## Data Availability

Not applicable.

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
