# Peer review of "Adaptive Mechanisms of Halophytes and Their Potential in Improving Salinity Tolerance in Plants"

_ijms, 2021, doi:10.3390/ijms221910733_

Round 1

Reviewer 1 Report

The review is devoted to the study of information on various halophytes that use different mechanisms of adaptation to the effects of salt.
According to the authors, such plants can be used as phytoremediation for saline soils, although this remains a controversial issue and there is no  soil phytoremediation technology.

The authors analyzed various aspects of the resistance of halophytophs; Of particular interest is the section devoted to the molecular mechanisms of plant adaptation.
Also, an important section is devoted to the study of various genes activated by salinity and cloned in other plants.

The article contains a sufficient number of literary sources, most of which are the most recent.

Pictures are clear, English is acceptable in my opinion.

The MS makes a good impression.

Author Response

Thank you very much for your positive evaluation of our manuscript. 

Reviewer 2 Report

This is a well written review on the "Adaptive mechanisms of halophytes and their potential in improving salinity tolerance in plants". Although pretty well written some minor English changes, particularly at grammar level.

As for the review itself, the Authors described most of the "hot topic" related to salinity and halophytes. I'd suggest to add a paragraph about roots adaptions. Lot of research has been conducted by several authors on root apoplastic barriers, the role of suberin and lignin, and the potential role of suberin lamellae in counteracting salt intrusion at endodermal level.

See the following links for more info about root apoplastic barriers:

  • Root vacuolar sequestration and suberization are prominent responses of Pistacia spp. rootstocks during salinity stress
  • Effects of soil salinity on citrus rootstock 'US-942'physiology and anatomy
  • Salt tolerance and exclusion in the mangrove plant Avicennia marina in relation to root apoplastic barriers

Author Response

We thankfully acknowledge your overall positive comment on our manuscript. 

Round 2

Reviewer 2 Report

The manuscript is now ready for publication.